# Clinical Impact of Immune Checkpoint Inhibitor (ICI) Response, DNA Damage Repair (DDR) Gene Mutations and Immune-Cell Infiltration in Metastatic Melanoma Subtypes

**DOI:** 10.3390/medsci10020026

**Published:** 2022-05-24

**Authors:** Charlotte Andrieu, Niamh McNamee, Anne-Marie Larkin, Alanna Maguire, Roopika Menon, Judith Mueller-Eisert, Noel Horgan, Susan Kennedy, Giuseppe Gullo, John Crown, Naomi Walsh

**Affiliations:** 1National Institute for Cellular Biotechnology, School of Biotechnology, Dublin City University, Glasnevin, D09 E432 Dublin, Ireland; charlotte.andrieu@dcu.ie (C.A.); nmcnamee@tcd.ie (N.M.); larkin.annemarie@itsligo.ie (A.-M.L.); alannamaguire@gmail.com (A.M.); 2Department of Life Sciences, Institute of Technology Sligo, F91 YW50 Sligo, Ireland; 3Siemens Healthcare Diagnostics Products GmbH, 50667 Cologne, Germany; mroopika@gmail.com (R.M.); judith.mueller-eisert@siemens-healthineers.com (J.M.-E.); 4Royal Victoria Eye and Ear Hospital, Adelaide Road, D02 XK51 Dublin, Ireland; noelhorgan@gmail.com (N.H.); skennedy@svhg.ie (S.K.); 5Department of Medical Oncology, St. Vincent’s University Hospital, D04 T6F4 Dublin, Ireland; giuseppe.gullo@hotmail.it (G.G.); john.crown@cancertrials.ie (J.C.)

**Keywords:** cutaneous melanoma, uveal melanoma, head and neck melanoma, mucosal melanoma, acral lentiginous melanoma, immunotherapy, genomics, targeted sequencing, predictive biomarkers, precision medicine

## Abstract

Molecular and histopathological analysis of melanoma subtypes has revealed distinct epidemiological, genetic, and clinical features. However, immunotherapy for advanced metastatic melanoma patients does not differ based on subtype. Response to immune checkpoint inhibitors (ICI) has been shown to vary, therefore, predictive biomarkers are needed in the design of precision treatments. Targeted sequencing and histopathological analysis (CD8 and CD20 immunohistochemistry) were performed on subtypes of metastatic melanoma (cutaneous melanoma (CM, *n* = 10); head and neck melanoma (HNM, *n* = 7); uveal melanoma (UM, *n* = 4); acral lentiginous melanoma (AM, *n* = 1) and mucosal melanoma (MM, *n* = 1) treated with ICI). Progression-free survival (PFS) was significantly associated with high CD8 expression (*p* = 0.025) and mutations in DNA damage repair (DDR) pathway genes (*p* = 0.012) in all subtypes but not with CD20 expression. Our study identified that immune cell infiltration and DDR gene mutations may have an impact in response to ICI treatment in metastatic melanoma but differs among subtypes. Therefore, a comprehensive understanding of the immune infiltration cells’ role and DDR gene mutations in metastatic melanoma may identify prognostic biomarkers.

## 1. Introduction

Immune checkpoint inhibitor (ICI) therapies which target the programmed cell death protein 1 (PD-1)/programmed death ligand 1 (PD-L1) or the cytotoxic T-lymphocyte-associated protein 4 (CTLA-4) alone or in combination are used in the treatment of multiple advanced solid tumours including metastatic melanoma. ICI has changed the treatment landscape for metastatic cutaneous melanoma (CM), with ICI combinations displaying objective response rates (ORR) of greater than 60%; moreover, compared to traditional chemotherapy, patients can achieve complete response and long-term survival with novel combination strategies of ICI [1,2,3].

Rare subtypes of non-cutaneous melanoma such as acral, mucosal and uveal melanomas have different clinical, histopathological and genomic features [4,5]. Acral lentiginous melanomas (AM) tend to arise on the non-hair skin areas such as the palms of the hands and soles of the feet; mucosal melanomas (MM) arise from melanocytes of the mucosal epithelium of the respiratory, alimentary and genitourinary tract regions; and uveal melanomas (UM) arise from melanocytes in the iris, ciliary body, or choroid [6]. Head and neck cutaneous melanomas (HNM) are commonly found on the occipital scalp and skin of the cheek, and represent a distinct entity of skin cancer [7,8,9]. Despite the advances in CM treatment and management, ICI therapy has shown varied and limited benefits for other subtypes of metastatic melanoma. These subtypes of metastatic melanoma tend to be less susceptible to ICI therapies compared to CM, potentially due to the low presence of tumour-infiltrating lymphocytes, low somatic mutational burden, and the lack of a UV-mutational signature [10]. Nevertheless, post-hoc analysis of clinical trials and small retrospective studies have shown disparities in response rates to ICI [11,12]. A further retrospective study of 428 metastatic melanoma patients treated with ICI observed the median OS as 45 months for CM, 17 months for AM, 18 months for MM, and 12 months for UM. Although long-term survival was observed, complete responses were rare in metastatic melanoma subtypes [13]. These studies suggest that ICI is a valuable treatment option for metastatic melanoma subtypes, but attention is needed in considering the predictive clinical and biological characteristics to identify patients who will benefit most from ICI. Clinical predictors of ICI response, such as tumour mutational burden (TMB), DNA damage response pathways, neo-antigen load, and tumour immune microenvironment (TME) biomarkers have been extensively studied [reviewed in Bai et al. [14]]. However, predictive biomarkers of ICIs efficacy involve complex interactions between the different subtypes of metastatic melanoma and the regulation of the immune system network. Therefore, we performed targeted exome sequencing and assessed the expression of TILs (CD8+) and tumour infiltrating B-cells (CD20+) in subtypes of 23 metastatic melanoma patients treated with ICI in order to explore their clinical impact as prognostic factors.

## 2. Materials and Methods

### 2.1. Patient Characteristics

This is a retrospective clinical study consisting of 23 patients diagnosed with metastatic melanoma and treated with ICI in St. Vincent’s University Hospital, Dublin between 2014–2017. The median age was 62 years (range 42–83). The 23 metastatic melanoma subtypes consisted of primary tumours derived from cutaneous melanoma (CM, *n* = 10); head and neck melanoma (HNM, *n* = 7); uveal melanoma (UM, *n* = 4); acral lentiginous (AM, *n* = 1) and mucosal melanoma (MM, *n* = 1). All 23 patients were treated with ICI (pembrolizumab, ipilimumab or nivolumab), 12/23 (52%) with one ICI and 11/23 (48%) with two or more ICIs. The majority of patients did not receive chemotherapy/targeted therapy 17/23 (74%) (Appendix A). This study was approved by the Institutional Review Board/Ethics Committee of St. Vincent’s University Hospital, Dublin. 

### 2.2. Immunohistochemistry Staining and Scoring

Representative 4 μm sections of formalin-fixed paraffin-embedded metastatic melanoma tumour tissues were cut using a microtome, mounted onto poly-l-lysine coated slides and dried overnight at 37 °C. Slides were stored at room temperature until required. Immunohistochemistry (IHC) staining was performed using an automated staining apparatus for IHC (Autostainer, DakoCytomation) according to the manufacturer’s guidelines. Optimum primary antibody dilutions were predetermined using known positive control tissues. Negative and known positive control sections were included in each run. Deparaffinisation and heat-induced epitope antigen retrieval (HIER) consisted of 40-min incubation in pH 9.0 buffer (Target Retrieval, DakoCytomation) in a 95 °C water bath followed by cooling to room temperature. In the Dako Autostainer, sections were treated with 3% H_2_O_2_ for 10 min to quench endogenous peroxidase and then rinsed. Quenched sections were incubated with antibodies, CD8 (clone C8/144B; Dako, Agilent, Santa Clara, CA, USA) and CD20cy (clone L26; Dako) for 30 min, followed by incubation with Dako REAL™ detection system, alkaline phosphatase/RED reagent for 30 min. The antigen-antibody complex was visualised using AP-Fast Red-type chromogen system. Sections were then counterstained with Dako REAL haematoxylin (S3301, Dako), and mounted using VectaMount AQ aqueous mounting medium (h-5501-60). Slides were reviewed by light microscopy. The presence of CD8+ TIL was scored as 0 (negative), low (1, positive cells at edge of tumour, <5% of area), moderate (2, tumour infiltration of TILs, 5–50% of area), or high (3, strong diffuse infiltration by TIL, >50% of area). Scores of 0, 1, 2 were collated and compared against score 3 for evaluation purposes. 

### 2.3. Targeted Sequencing and Analysis

Retrospectively, formalin-fixed paraffin-embedded (FFPE) tissue was analysed by using a comprehensive hybrid capture–based next-generation sequencing assay as previously described [15]. Genomic DNA was extracted from FFPE, sheared (Covaris, Woburn, MA, USA), and subjected to hybrid capture–based next-generation sequencing to detect point mutations, small insertions and deletions, copy number alterations and rearrangement/gene fusions in a single assay. In brief, after shearing, adapters were ligated and individual genomic regions of interest were enriched using complementary bait sequences (hybrid capture procedure). The selected baits ensure optimal coverage of all relevant genomic regions. After enrichment, targeted fragments were amplified (clonal amplification) and sequenced in parallel at high sequencing depth. Reads were trimmed and aligned to the hg38 reference genome using BWA and duplicate reads were marked. Base recalibration was conducted with GATK. Variant calling was performed using Mutect2 from GATK (4.1.3). The Copy Number Aberrations (CNA) burden was estimated for each sample and enrichment analysis was performed using EnrichR [16,17,18].

### 2.4. Statistical Analysis

Progression-free survival (PFS) was calculated from the start date of ICI for metastatic disease to systemic relapse/death. Overall survival (OS) was calculated from date of diagnosis for metastatic disease to death or latest follow-up. Comparisons between groups were performed using Fisher’s exact test. A *p*-value < 0.05 was considered statistically significant. Univariate and Cox regression analysis was used to determine independent prognostic predictors of PFS and OS. The Kaplan–Meier survival estimator method was applied to calculate PFS and OS, and the log-rank test was used for assessment of statistical significance. Statistical analysis was performed using STATA v17.0 (StataCorp, LLC, College Station, TX, USA). 

## 3. Results

### 3.1. IHC Expression of CD8 and CD20 in Metastatic Melanoma Subtypes

Of the 23 patients, only 22 had available specimens for IHC (one UM subtype contained too much melanin for analysis). Among the 22 patients, 13/22 (59%) displayed low IHC staining for CD8 with peritumour and/or interstitial TIL infiltration (score 0, 1, 2) and 9/22 (41%) displayed high CD8 staining with strong diffuse immune cell infiltration (score 3). CD20 low IHC staining (score 0, 1, 2) was observed in 15/22 (68%) of tumours and 7/22 (32%) displayed (score 3) high staining (Figure 1A–F). The majority of CM and HNM patients displayed low CD8 and CD20 staining, compared to UM subtype samples which displayed high CD8 and CD20 staining (Figure 1G–H). There was no significant correlation with CD8 expression and clinical variables, whereas CD20 high expression was significantly associated with UM (*p* = 0.023) (Table 1).

### 3.2. CD8 and CD20 Cell Infiltration Score Associated with Progression-Free Survival (PFS) in Metastatic Melanoma Subtypes

In the entire patient group (*n* = 23), the median OS and PFS times were 27.72 months (95% CI: 1.81–177.74) and 3.35 months (95% CI: 0.36–39.65), respectively. A high CD8 score was significantly associated with better progression-free survival compared to a low CD8 score (median PFS, 6.93 months (95% CI: 2.75–19.44) vs. 2.77 months (95% CI: 0.36–7.45), *p* = 0.025, log rank) (Figure 2A). CD20 B-cell infiltration score did not significantly improve PFS (median PFS, 3.07 months (95% CI: 0.36–16.65) vs. 3.41 months (95% CI: 0.45–19.44), *p* = NS, log rank) in metastatic melanoma (Figure 2B).

### 3.3. Impact of DNA Damage Repair (DDR) Gene Mutation on Clinical Outcome

We next investigated the impact of the number of DDR gene mutations on the clinical outcome of metastatic melanoma subtypes after ICI therapy. We found that 83% of metastatic melanoma cases had at least one mutation in a DDR pathway gene (Figure 3A). DDR gene mutations were significantly associated with better PFS (3.54 months (95% CI: 2.75–9.10) vs. 2.75 months (95% CI: 0.36–3.35), *p* = 0.012, log rank) in the full cohort (*n* = 23) (Figure 3B). 

There was a significant association observed with DDR gene mutation and age (*p* = 0.012, Fisher’s exact test), but no significant correlation with CD8/CD20 expression and clinical variables (Table 1).

### 3.4. Copy Number Aberrations (CNA) of Metastatic Melanoma Subtypes and Clinical Impact

The median CNA count for the full metastatic melanoma cohort was 168 (range: 26–1402). Sub-analyses based on subtype showed the median CNA count in CM, 202.5 (range: 26–639); HNM, 147 (range: 32–547); UM, 308.5 (range: 168–1402); AM/MM, 151 (range: 145–157). UM has the highest CNA count compared to the other tumours, but not significantly (Figure 4A). Metastatic melanoma tumours with a high CNA count showed better PFS (HR: 0.39; median PFS: 5.25 months vs 3.54 months; *p* = 0.058, log rank) (Figure 4B). 

Univariate analyses for PFS and OS are shown in Table 2. Univariate analysis revealed that high CD8 expression (HR 0.32; 95% CI: 0.11–0.93, *p* = 0.039) and the presence of DDR gene mutations (HR 0.28; 95% CI: 0.10–0.81, *p* = 0.019) are favourable prognostic factors for PFS; no variable was found to be a prognostics factor for OS. The variables remained as independent favourable prognostic factors in multivariate analysis (*p* = 0.018).

## 4. Discussion

Clinical markers of response to ICI therapies have been extensively studied. Comprehensive predictive and prognostic models have been developed integrating the expression of intermolecular interactions within the tumour, its microenvironment as well as the correlation with tumour genome mutational, neo-antigen burden, and genetic variations in DNA mismatch repair genes [14]. However, conflicting evidence into the clinical predictive efficacy of ICI biomarkers in different cancer subtypes exists [19]. PD-L1 expression measured by immunohistochemistry (IHC) has emerged as a widely used biomarker of response to ICI; however, patients with PD-L1 negative tumours also have shown efficacy to ICI treatment. In a clinical trial of refractory or metastatic cervical cancer patients treated with nivolumab, the overall response rates (ORR) were 2/10 (20%) in PD-L1 positive patients and 1/6 (16.7%) in PD-L1 negative patients. Indeed, a patient with a PD-L1 negative tumour had a durable partial response that exceeded 24 months [20].

Responses in other cancer types such as small cell lung cancer (SCLC), squamous cell carcinoma of the head and neck (SCCHN) and cutaneous melanoma led to an accelerated all-comer approval for pembrolizumab regardless of PD-L1 status [21,22]. However, a recent follow-up of the cervical cancer KEYNOTE-158 study found that all 14 responses were in patients with PD-L1–positive tumours (ORR 17.1%) and with no responses observed in the PD-L1–negative cohort (ORR 0.0%) [23]. 

Metastatic melanoma is a heterogeneous group of cancers characterised by site of origin, subtypes based on the cumulative levels of exposure to ultraviolet (UV) radiation [10,11,24]. Melanomas found in sun damaged areas of the body tend to have a higher mutational burden than tumours arising from non-sun exposed areas correlating with somatic mutational profile of the tumour and high response rates after ICI therapy [25]. In addition, many studies have found that melanoma is characterised by high immunogenicity and patients with high TIL cells have shown favourable therapeutic outcomes and prognosis. However, as ICI only shows efficiency in some metastatic melanoma patients (majority of cutaneous melanoma), this study examined the clinical relevance of CD8+ TIL and CD20+ tumour infiltrating B cells and the genomic mutational landscape in different subtypes of 23 metastatic melanoma patients treated with immune checkpoint inhibitors (ICI). TILs have previously been identified as prognostic and predictive biomarkers in many cancers, including melanoma. However, complex tumour-immune interactions exist whereby some cancers with similar TIL cell expression respond differently to immunotherapy [26]. We observed lower CD8+ and CD20+ expression in subtypes CM and HNM. CD8+ TILs have been detected in metastatic uveal melanoma (MUM) [27], however, no difference in CD8+ infiltrating T cells between metastatic CM and MUM was observed [28]. However, the number of UM in our study was very low (*n* = 4). Location of the tumour in head and neck melanoma has a significant impact on prognosis, with the scalp displaying the worst prognosis, followed by the ear, cheek, and neck [9]. A rare variant of HNM is desmoplastic melanoma (DM), which is associated with old age, chronic sun exposure, and location on the head and neck. HNM DM has shown a great response rate to ICI therapy [29]. However, Frydenlund et al. [30] found that the presence of CD8+ lymphocytes correlated significantly with depth of invasion > 1 mm, and PD-L1 expression in DM. Additionally, CD8+ TILs correlated with PD-L1 expression, which was associated with tumour aggressiveness and progression in DM [31]. However, in our study, HNM did not harbour a higher mutational burden comparable to other metastatic melanoma subtypes. High tumour mutation burden (TMB) has been associated with response to ICI in several cancers. Several studies of multiple cancers have shown that ICI mean response rate positively correlates with TMB [32]. The FDA have approved TMB (TMB-High defined as ≥10 mutations/megabase of DNA (mut/Mb), as determined by the targeted sequencing FoundationOne CDx (F1CDx) assay) as a companion diagnostic biomarker for pembrolizumab [33]. However, some clinical studies also showed that high TMB does not predict clinically relevant responses to ICI in all cancer types. Specifically, cancer types such as breast, prostate and glioma displayed no relationship between high CD8 TIL expression and neo-antigen load. Designated TMB-H tumours failed to achieve a 20% ORR (15.3%) and displayed significantly lower ORR relative to TMB-Low tumours [34]. 

In agreement with many studies, DDR gene mutations correlated with better PFS in the metastatic melanoma cohort. Additionally, metastatic melanoma tumours with a high CNA count showed better PFS (HR: 0.39; median PFS: 5.25 months vs. 2.92 months; *p* = 0.058, logrank) but this was not significant. In univariate analysis DDR gene mutations and CD8 high were prognostic indicators of better PFS. Therefore, this study highlights the differential expression of TILs and genomic variation (DDR gene mutations and CNV counts) in subsets of metastatic melanoma. The variability of these biomarkers highlights the need for additional predictive and prognostic specific to metastatic melanoma subtypes. However, this study is retrospective in nature, and is limited by the small sample and subtype size, therefore further analysis on larger cohorts would be required.

## Figures and Tables

**Figure 1 medsci-10-00026-f001:**
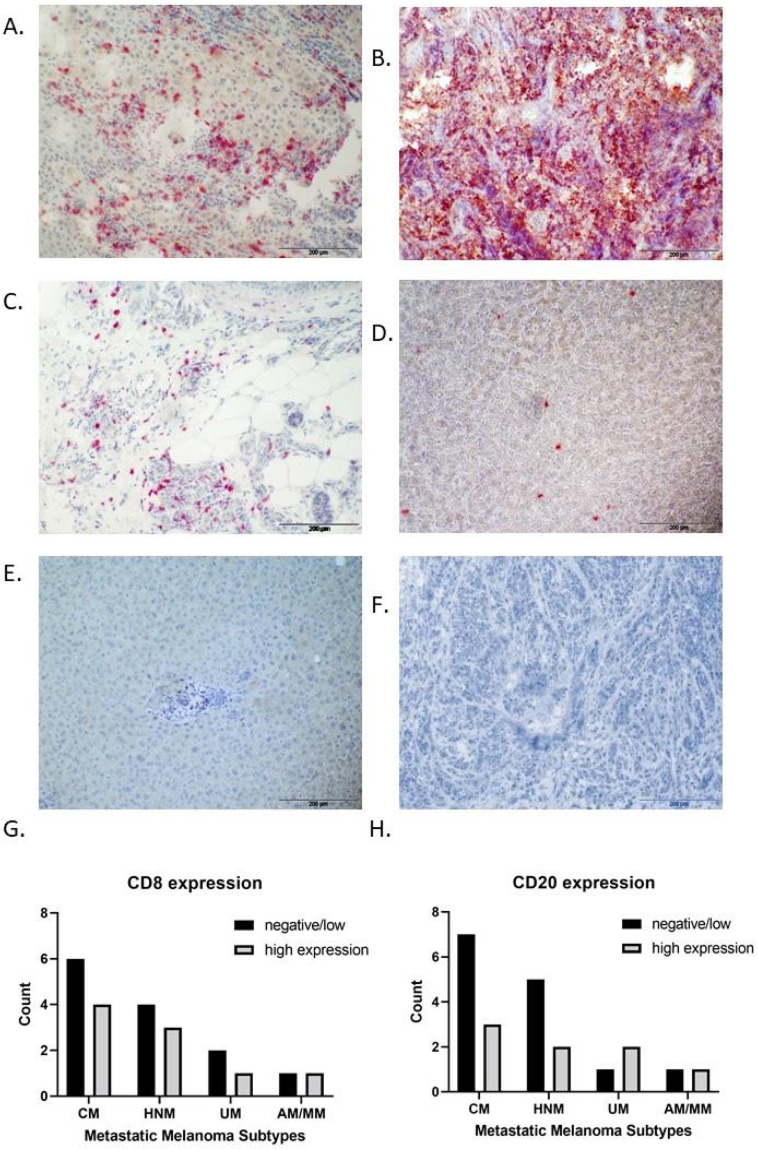
Immunohistochemical staining of tumour samples at ×200 magnification. Representative high expression (strong diffuse immune cell infiltration) for (**A**). CD8 in HNM, (**B**). CD20 staining in HNM. Representative low expression (peritumour and/or interstitial immune infiltration) for (**C**). CD8 in MM, (**D**). CD20 expression in CM. Representative negative expression for (**E**). CD8 and (**F**). CD20 expression in HNM. Overall expression of (**G**). CD8 and (**H**). CD20 in metastatic melanoma subtypes.

**Figure 2 medsci-10-00026-f002:**
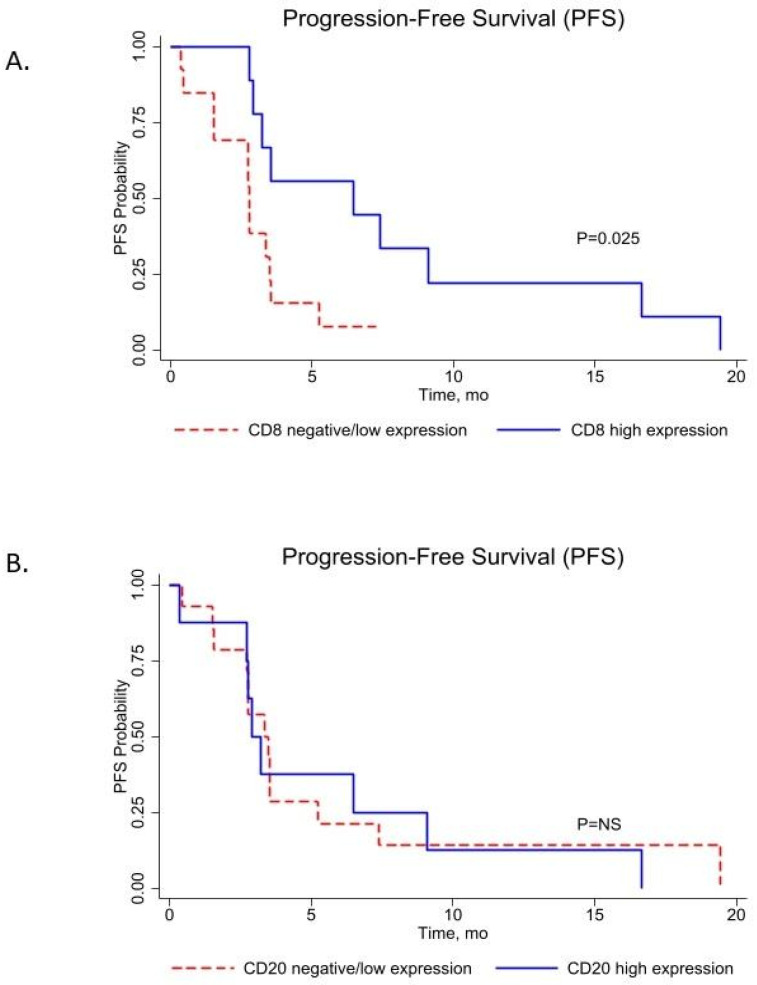
Kaplan–Meier plots of progression-free survival (PFS) between (**A**). CD8 high and CD8 low, and (**B**). CD20 high and CD20 low TIL score groups with log-rank *p* value.

**Figure 3 medsci-10-00026-f003:**
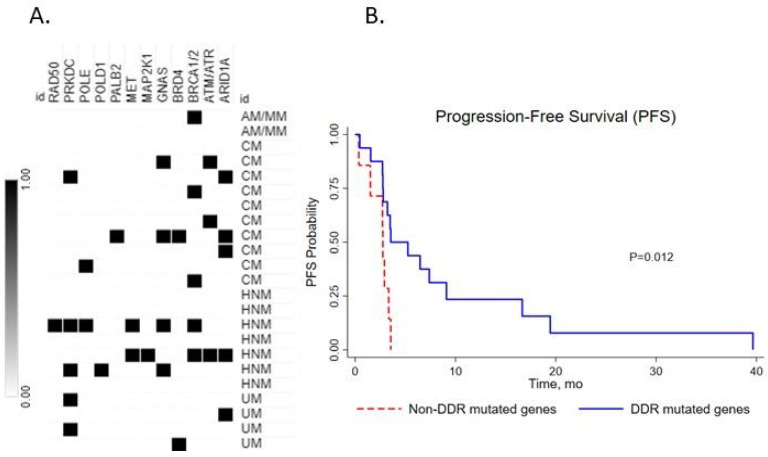
(**A**) Map of metastatic melanoma subtypes with DDR gene mutations. (**B**) Kaplan–Meier plots of progression-free survival (PFS) for cohorts with DDR mutated genes and without DDR mutated genes with log-rank *p*-value.

**Figure 4 medsci-10-00026-f004:**
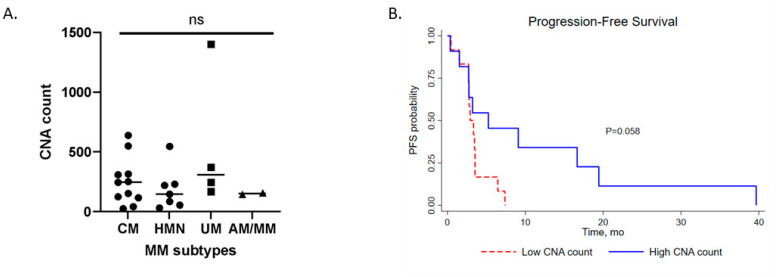
(**A**). Profile of CNA count on subtypes of metastatic melanoma (CM

; HNM

; UM
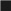
; AM/MM
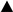
). (**B**). Progression-free survival (PFS) of CNA low (red line) and high (blue line) in the total metastatic melanoma cohort. Kaplan–Meier survival curves with a log-rank test were used for the analysis. ns, not significant.

**Table 1 medsci-10-00026-t001:** Association between the expression of CD8, CD20 expression, DDR gene mutations, CNA count and clinical features.

		**CD8 (*n* = 22)**			**CD20 (*n* = 22)**			**DDR Mutation (*n* = 23)**		
		**Neg/Low**	**High**	***p*-Value**	**Neg/Low**	**High**	***p*-Value**	**Neg**	**Pos**	***p*-Value**
		**(*n* = 14)**	**(*n* = 8)**		**(*n* = 14)**	**(*n* = 8)**		**(*n* = 7)**	**(*n* = 16)**	
**Age**	<62	5	5	0.221	4	6	0.048	6	4	0.012
	≥62	9	3		10	2		1	12	
**Melanoma subtype**	CM	7	3	0.917	7	3	0.720	2	8	0.171
	HNM	4	3		5	2		4	3	
	UM	2	1		1	2		0	4	
	AM/MM	1	1		1	1		1	1	
**Chemotherapy**	Yes	2	4	0.096	3	3	0.369	1	5	0.382
	No	12	4		11	5		6	11	
**Immunotherapy**	1 ICI	7	4	0.670	7	4	0.670	5	7	0.222
	2+ ICI	7	4		7	4		2	9	
**CD20**	High	3	5	0.072				3	5	0.510
	Neg/Low	11	3					4	10	
**DDR mutation**	Positive	9	6	0.490						
	Negative	5	2							
**CNA count**	High	5	5	0.221	5	5	0.221	3	8	0.556
	Low	9	3		9	3		4	8	

**Table 2 medsci-10-00026-t002:** Univariate Cox Regression analysis of clinical variables associated with PFS and OS.

			**PFS**	**Univariate Analysis**		**OS**	**Univariate Analysis**	
			**Median PFS, Months (Range)**	**HR (95% CI)**	***p*-Value**	**Median OS, Months (Range)**	**HR (95% CI)**	***p*-Value**
			**3.35 (0.36–39.65)**			**27.72 (1.80–177.74)**		
**Categories (*n* = 23)**								
**Age**	<62	10	3.23 (0.36–16.65)	Ref		44.13 (5.12–100.66)	Ref	
	≥62	13	3.35 (0.45–39.65)	0.74 (0.30–1.78)	0.502	27.36 (1.80–177.74)	1.27 (0.415–3.92)	0.670
**Melanoma Subtype**	CM	10	3.54 (0.36–19.45)	Ref		51.51 (1.8–100.67)	Ref	
	HNM	7	2.92 (1.51–7.46)	1.38 (0.48–3.98)	0.548	27.73 (5.12–177.74)	1.66 (0.36–7.66)	0.513
	UM	4	3.01 (1.54–39.65)	0.82 (0.22–3.07)	0.777	19.52 (10.74–50.13)	4.86 (1.12–21.05)	0.034
	AM/MM	2	5.05 (2.72–7.39)	1.27 (0.27–6.01)	0.756	20.90 (14.06–27.72)	5.23 (0.86–31.73)	0.072
**Chemo-Therapy**	Yes	6	4.40 (2.92–7.39)	Ref		38.04 (23.52–100.66)	Ref	
	No	17	2.79 (0.36–39.65)	1.00 (0.37–2.69)	0.997	27.30 (1.80–177.74)	0.69 (1.87–2.55)	0.582
**Immuno-Therapy**	1 ICI	12	2.85 (0.36–39.65)	Ref		25.74 (1.80–80.91)	Ref	
	2+ ICI	11	3.48 (2.72–19.44)	0.99 (0.41–2.35)	0.984	35.58 (14.06–177.74)	0.36 (0.11–1.20)	0.098
**DDR Mutation**	Neg	16	2.75 (0.36–3.54)	Ref		40.50 (5.12–177.74)	Ref	
	Pos	7	4.40 (0.45–39.65)	0.28 (0.10–0.81)	0.019	27.54 (1.80–100.66)	0.98 (0.29–3.25)	0.984
**CNA Count**	Low	12	2.92 (1.51–3.54)	Ref		31.65 (1.80–177.74)	Ref	
	High	11	5.25 (1.54–19.4)	0.39 (0.14–1.08)	0.070	23.75 (5.12–100.66)	1.24 (0.417–3.74)	0.690
**Categories (*n* = 22)**								
**CD8 Expression**	Neg/Low	14	2.78 (0.36–7.45)	Ref		27.72 (1.81–177.74)	Ref	
	High	8	6.93 (2.75–19.44)	0.32 (0.11–0.94)	0.039	34.11 (5.12–53.61)	0.83 (0.24–2.81)	0.773
**CD20 Expression**	Neg/Low	14	3.41 (0.45–19.44)	Ref		27.54 (1.81–177.74)	Ref	
	High	8	3.07 (0.36–16.65)	1.03 (0.41–2.57)	0.935	32.01 (10.74–80.91)	0.91 (0.27–3.07)	0.888

## Data Availability

The sequencing and clinical variables for this study are available from EGA European Genome-Phenome Archive accession number: EGAD00001008336.

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
