# Peer review of "Clinical Impact of Immune Checkpoint Inhibitor (ICI) Response, DNA Damage Repair (DDR) Gene Mutations and Immune-Cell Infiltration in Metastatic Melanoma Subtypes"

_medsci, 2022, doi:10.3390/medsci10020026_

Round 1

Reviewer 1 Report

Review

Clinical impact of immune checkpoint inhibitor (ICI) response, 2 DNA damage repair (DDR) gene mutations and immune-cell 3 infiltration in subtypes of metastatic melanoma

By Charlotte Andrieu1...et al.

studied five metastatic melanoma entities using targeted sequencing and histopathological analysis regarding the effect of ICI. Overall, they found better prognosis for genomic deficient cases (mutations in DDR genes) as expected. Anyway, these results are important as reported fact, particularly regarding the heterogeneous appearance of melanomas.

The study is sound and well done (selection nd reporting of cases, statistcal analysis, figures). The discussion is instructive also for non-clinical specialists like me. The paper well fits into the scope of the journal.

I have only three minor points:

  • Please spell out CNA in the methods section.
  • Table 1: “Correlation”, I suspect, is not the right terminus. One espects quantification in terms of correlation coefficients or so... Might “assiciation”, “relation”, relatedness”, “comparison between” or whatever is better suited.
  • Its a bit major issue: Overall the sample size is rather small. Even if statistics is well done, the “error of the error” matters, meaning formal significance (eg p<0.05) is uncertain to a certain degree. Hence, the reported results have to interpreted with coution especially because of the heterogeneity of the tumour entities. Might the authors want to address this shortly in the discussion section.

Author Response

Response to Reviewer 1 Comments:

  • Please spell out CNA in the methods section.

Response: The methods section has been updated with the Copy Number Aberration (CNA) burden spelled out.

  • Table 1: “Correlation”, I suspect, is not the right terminus. One espects quantification in terms of correlation coefficients or so... Might “assiciation”, “relation”, relatedness”, “comparison between” or whatever is better suited.

Response:The table 1 description has been changed; “association” is indeed better suited in that case.

  • Its a bit major issue: Overall the sample size is rather small. Even if statistics is well done, the “error of the error” matters, meaning formal significance (eg p<0.05) is uncertain to a certain degree. Hence, the reported results have to interpreted with coution especially because of the heterogeneity of the tumour entities. Might the authors want to address this shortly in the discussion section.

Response: The authors accept the limitations of the study specifically regarding the small sample subtype cohort and therefore, have removed specific descriptions/analyses of the stratification subtypes in Figures and highlight the limitations of the study in the discussion.

Reviewer 2 Report

The research paper analyses the effectiveness of ICI (checkpoint inhibitors) therapy in distinct groups of metastatic melanoma patients. The most common type of melanoma, cutaneous melanoma has strong indications to be treated with ICIs, however it’s still unknown if the same approach would be equally effective in other metastatic melanoma subtypes.

Main points:

  1. The weakest point of the study is the small number of patients with rare forms of metastatic melanoma. Furthermore, head and neck melanoma (HNM) is often classified as cutaneous melanoma (CM), but with specific location. The values obtained for analysed parameters (CD staining and DDR count) are similar in “CM” and “HNM” cohort containing 10 and 7 patients, respectively. On the opposite are 3 samples of uveal melanoma, 1 of acral lentiginous melanoma and 1 of mucosal melanoma. Therefore, due to the low number of samples from patients with rare melanoma subtypes, no clear conclusion for subtypes other than CM/HNM can be withdrawn here. And the results shown for CM lack novelty, as the TIL’s infiltration and high TMB (tumour mutational burden) are already known as good predictors of immunotherapy efficiency. The added value of showing molecular characteristics dependent ICI efficiency in rare metastatic melanoma subtypes is missing.

Minor points:

  1. Materials and Methods section contains “copy-paste” passage from the MDPI instructions for authors.
  2. The particular mutations in DDR genes are not revealed– please provide the list of mutations, and information on the type of mutation (type of mutation and effect on protein activity or expression, if known). Here, DDR gene mutations count are treated as an equivalent of TBM (tumour mutational burden), therefore only the pinpointing of inactivating mutations would make this approach reasonable. Moreover, the description of DNA sequencing implies that broader analysis of mutational load in the studied samples is feasible.
  3. The sequencing method has not been sufficiently described, e.g. the technical data are missing, like what type of sequencing machine was used, what kind of library prepared etc. Are the raw sequencing data uploaded and made available for interested parties?
  4. Figure 1A. legend. As I understand, the intention was to show the examples of high-low level of staining for CD8 or CD20. However, it would be more appropriate and in accordance with good IHC practice if the origin (e.g. type of melanoma) of tissue slices shown on A-F panels is uncovered.
  5. The information on patients, samples and treatment is scarce. How the tissue specimens were obtained? Was it from primary tumour site or metastasis? The information on age/ melanoma subtype/ PFS/OS/treatment of each patient needs to be provided in the separate table. The data from Table 3 suggest that some patients lived longer than 14 years post diagnosis (177 months). Is it true?
  6. Table 1. It’s unclear if p-value for DDR (=0.012) refers to patients younger than 62y only or all patients. Moreover, there are 10 younger patients and 16 older patients, no way to add up and get 23 – please correct these records!
  7. Line 71: “…in 23 subtypes of metastatic melanoma patients…” please correct
  8. The characteristic “non-DDR mutated genes” is quite unfortunate, as it suggests that some genes non related to DDR are mutated? It requires rephrasing, like into “without DDR genes mutation” in opposite to “with DDR genes mutations”.
  9. Editing problem with Table 2. Table should be tidied up, probably the orientation change from a portrait to landscape may help. Now, it’s hard to find “what is what” in this table and draw any conclusion.

Round 2

Reviewer 2 Report

There are two versions of Figure 3 and Figure 4 each in the revised manuscript. Please correct, in line with the figure legend.